# Designing appropriate, acceptable and feasible community-engagement approaches to improve routine immunisation outcomes in low- and middle-income countries: A synthesis of 3ie-supported formative evaluations

**Stuti Tripathi**👤*, **Monica Jain, Avantika Bagai, Kirthi V. Rao**

International Initiative for Impact Evaluation (3ie), New Delhi, India

* stutitripathi@gmail.com

## Abstract

### Introduction

As global child vaccination coverage has plateaued, understanding how to increase routine child vaccination rates further is key to avoiding preventable disease and death. To analyse how community engagement strategies can increase child vaccination, we synthesise the results from formative evaluations of interventions that aimed to increase vaccination coverage in Ethiopia, Myanmar, Nigeria, and Pakistan

### Methods

This paper uses an inductive qualitative approach to synthesise the results from the six evaluations, gathering lessons for designing context appropriate interventions that are feasible to implement and acceptable to providers, communities, and caregivers.

### Results

Assessing contextual, caregiver-level and provider-level barriers to vaccination is key to identifying appropriate engagement strategies. Across all contexts, low knowledge about the schedule of vaccines and the importance of timeliness represented a barrier to child immunisation. Despite the variability in how studies measured and reported caregiver attitudes, vaccine hesitancy was not found to represent an important barrier to immunisation. Frontline health workers played a critical role in community engagement approaches to increase vaccination. Interventions successfully obtained community buy-in by centre-staging community members, especially leaders, ensuring their participation in monitoring, and making immunisation an agenda item on community platforms. Interventions were implemented through existing health systems with substantial assistance from research teams. Limited data was available about intervention costs.

**Data Availability Statement:** On careful review of the manuscript, we are happy to confirm that we

only use findings discussed in the published study reports for the synthesis paper. The published reports can be found here: https://www.3ieimpact. org/our-work/immunization. Links to individual study reports can be found below: Fifth Child Ethiopia: https://www.3ieimpact.org/sites/default/ files/2019-03/FE02-TW10.1017-The-Fifth-Child-Project-Ethiopia-web.pdf Pastoralist Ethiopia: https://www.3ieimpact.org/sites/default/files/2019-03/FE03-TW10.1076-Immunisation-coverage-Ethiopia-web.pdf CCCI Myanmar: https://www. 3ieimpact.org/sites/default/files/2019-03/FE-TW10. 1117-Community-checklists-immunisation-Myanmar-web.pdf PAR Nigeria: https://www. 3ieimpact.org/sites/default/files/2019-03/FE-TW10. 1054-Increasing-immunisation-coverage-Nigeria. pdf VIR Nigeria: https://www.3ieimpact.org/sites/ default/files/2019-05/FE06-TW10.1113-VIR-band-Nigeria_0.pdf VIR Pakistan: https://www.3ieimpact. org/sites/default/files/2019-04/FE-TW10.1030-VIR-band-Pakistan.pdf Only information contained in lines 322-323, 328-329 and 332-335 (under section 3.3.1 on working with health systems) is drawn from other proprietary documents (such as grant application and study progress reports submitted by research teams to 3ie in confidence). These documents contain other sensitive and identifiable information that might not be appropriate to share more widely.

**Funding:** This work was made possible by a grant from the Bill & Melinda Gates Foundation to the International Initiative for Impact Evaluation (3ie). The grant number is OPP1115129. Website URL - https://www.gatesfoundation.org/ Under the grant conditions of the Foundation, a Creative Commons Attribution 4.0 Generic License has already been assigned to the Author Accepted Manuscript version that might arise from this submission. The funders had no role in study design, data collection and analysis, decision to publish, or preparation of the manuscript.

**Competing interests:** We have read the journal's policy and the authors of this manuscript have the following competing interests: Supported by the Bill and Melinda Gates Foundation, the International Initiative for Impact Evaluation (3ie) provided funding and technical assistance for the six formative evaluations included in this synthesis. 3ie's technical assistance included review of study designs, analysis plans, and data collection instruments; and ongoing quality assurance, including advising research teams on various study aspects such as challenges in study implementation and engagement with stakeholders to promote uptake and use of evidence. Authors MJ and AB were involved in providing research

## Conclusions

Interventions designed around community engagement strategies can be appropriate, acceptable, and feasible approaches to overcome barriers to vaccination in a variety of low- and middle-income country contexts. However, questions remain about the ability of health systems to implement interventions at scale, both from a cost perspective and a capacity perspective.

## 1. Introduction

Global child mortality has reduced by a quarter in the past decade given the major thrust to increase vaccination coverage [1]. Government expenditure on national immunisation programmes increased by one-third in low- and middle-income countries (L&MICs). While the number of National Immunization Technical Advisory Groups nearly tripled to 114 from a mere 41 in 2010 [1], vaccination rates for diphtheria, tetanus and pertussis (DPT) and measles vaccines have since plateaued at 85 per cent. In 2019, close to 20 million children did not receive the three recommended doses of DPT vaccine, often used as an indicator to assess countries' performance on routine immunisation [1]. In 2021, vaccination rates declined, and number of unvaccinated children rose as the COVID-19 pandemic strained health systems. Ten countries, including Nigeria, India, the Democratic Republic of Congo and Pakistan, account for two out of five unvaccinated children globally [2].

To achieve its vision in which "all individuals and communities enjoy lives free from vaccine-preventable diseases", the Global Vaccine Action Plan came up with a set of guiding principles–one of which emphasises the community's role and its shared responsibility with the government to help "individuals and communities understand the value of vaccines and demand immunisation as both their right and responsibility" [3]. The WHO, too, recognises community engagement as pivotal in addressing vaccine hesitancy and the issue of service quality to build 'people-centred resilient health services' [1].

### 1.1. Building evidence on community engagement approaches

A scoping study by International Initiative for Impact Evaluation (3ie) pointed to important knowledge gaps on the effectiveness of community engagement approaches in addressing social and behavioural barriers to uptake of immunisation services [4]. Drawing on this, 3ie commissioned an evidence programme in 2015 to generate rigorous evidence on the role of community engagement approaches in L&MICs in addressing issues around last-mile delivery of vaccination services and behavioural, social and practical constraints faced by caregivers.

Six 3ie-funded formative evaluations and seven impact evaluations looked at interventions that used various tools and approaches to encourage community engagement. While formative evaluations focussed on questions around feasibility, appropriateness and acceptability of such tools and approaches, the impact evaluations sought to assess their effectiveness in increasing immunisation uptake.

### 1.2. Study objective

This paper qualitatively synthesises findings from the six 3ie-supported formative evaluations undertaken in Ethiopia, Myanmar, Nigeria, and Pakistan [5–10] and reflects on learnings in designing context-appropriate and feasible interventions acceptable to all stakeholders. It

teams with technical assistance. However, procedural safeguards were put in place to mitigate any risk of conflicting or competing interest. First, the lead author of this study, ST, has not been involved in quality assurance of these formative evaluations. Second, the review was undertaken only after all formative evaluation reports were completed and made publicly available. Lastly, the authors have no financial interest in this area. All underlying data for the paper is drawn from publicly available reports of studies that have ethics board approvals. All reports can be accessed here: https://www.3ieimpact.org/our-work/immunization Information from other proprietary documents (such as grant application and progress reports) has been drawn in only three instances when discussing strong relationships between researchers and their implementing agency counterparts in Fifth Child Ethiopia and VIR Nigeria in section 3.3.1 on working with the health systems. These documents contain other sensitive and identifiable information that might not be appropriate to share more widely. We confirm that this does not alter our adherence to PLOS ONE policies on sharing data and materials.

further seeks to broaden the understanding of formative evaluations by discussing methodological and conceptual underpinnings of the approach and placing it in the wider implementation research literature.

The US Centers for Disease Control and Prevention defines formative evaluation as research conducted to ensure "a programme or activity is feasible, appropriate, and acceptable before it is fully implemented. It is usually conducted when a new programme or activity is being developed or when an existing one is being adapted or modified" [11]. It can help understand what participants think about the programme, the features or components that may or may not work [12] and make incremental improvements [13].

For programme feasibility, acceptability, and appropriateness, the paper draws on Peters et al. [14] definition of the implementation outcomes. Acceptability is defined as the perception among stakeholders (e.g., consumers, providers, managers, policymakers) that an intervention is agreeable. Feasibility is the extent to which an intervention can be carried out in a particular setting or organisation. Appropriateness is seen as the perceived fit or relevance of the intervention in a particular setting or for a particular target audience or problem.

## 2. Material and methods

We followed a method of inductive enquiry to synthesise individual study findings and learnings around key themes of intervention appropriateness, acceptability, and feasibility. The method can be described as a form of thematic synthesis [15]. Information was systematically reviewed and coded against the following primary themes for emerging patterns and trends:

i.   The intervention context and needs identification

ii.  Discussion and review of existing evidence

iii. Intervention and study details

iv.  Stakeholder engagement

v.   Study findings and implications for policy and programme

We included sub-themes to capture individual study nuances and adjusted these as new information emerged. The full coding framework is included in S1 Table.

We used Nvivo, a qualitative data analysis software, to code relevant documents. The information was mainly derived from documents provided to 3ie by study teams through the course of the study as part of an agreed deliverables and disbursement schedule. At the minimum, these consisted of baseline and endline reports and yearly progress reports on study implementation and stakeholder engagement. Study teams also shared copies of presentations made to stakeholders, meeting minutes and media coverage, if any.

The synthesis paper conforms to reporting guidelines as set out in the Standards for Reporting Qualitative Research [16]. It is based on secondary research, drawn from published reports that can be found on the 3ie website. Since the synthesis did not involve any primary interaction with human subjects or animals, approval from an Ethics Committee or Institutional Board was not required.

While authors Monica Jain and Avantika Bagai managed 3ie's evidence programme on immunisation and quality assured the studies, the research design, implementation and findings are those of the study teams and their intellectual property. Lead author Stuti Tripathi and fourth author Kirthi V. Rao were not involved in the management or supervision of the evidence programme.

## 2.1. Overview of the interventions

By engaging community in various ways, the interventions sought to tackle last-mile delivery issues and behavioural, social and practical constraints faced by caregivers. Almost all interventions included a component on health worker sensitisation and training and assigned key roles to community leaders in awareness-raising and health system monitoring to improve immunisation coverage. The descriptions of the interventions are provided in S2 Table.

## 2.2. Community engagement approaches: Theory of change and role of formative evaluations

The interventions varied in the tools and approaches they used to engage the community and the stage at which they involved the community. In their systematic review, Jain and colleagues [17] offer a theoretical framework (Fig 1) that captures this diversity in community engagement approaches and broadly indicates the causal links that undergird community engagement interventions, and the assumptions made along the way, to affect the behavioural, social, and practical drivers of immunisation and improve service delivery.

Formative evaluations helped interrogate theory of change assumptions by unpacking how aspects such as appropriateness, acceptability, and feasibility impinged upon intervention logic. Tumilowicz et al [18] talk about the concept of 'programme impact pathways' as critical in 'organising and describing' the various tasks and activities, and considerations that need careful attention when planning the intervention stages. These include identifying possible efficiencies and inefficiencies in the intervention flow along the impact pathway and identifying the desirable stakeholder behaviour (provider and caregiver) at each step of the intervention.

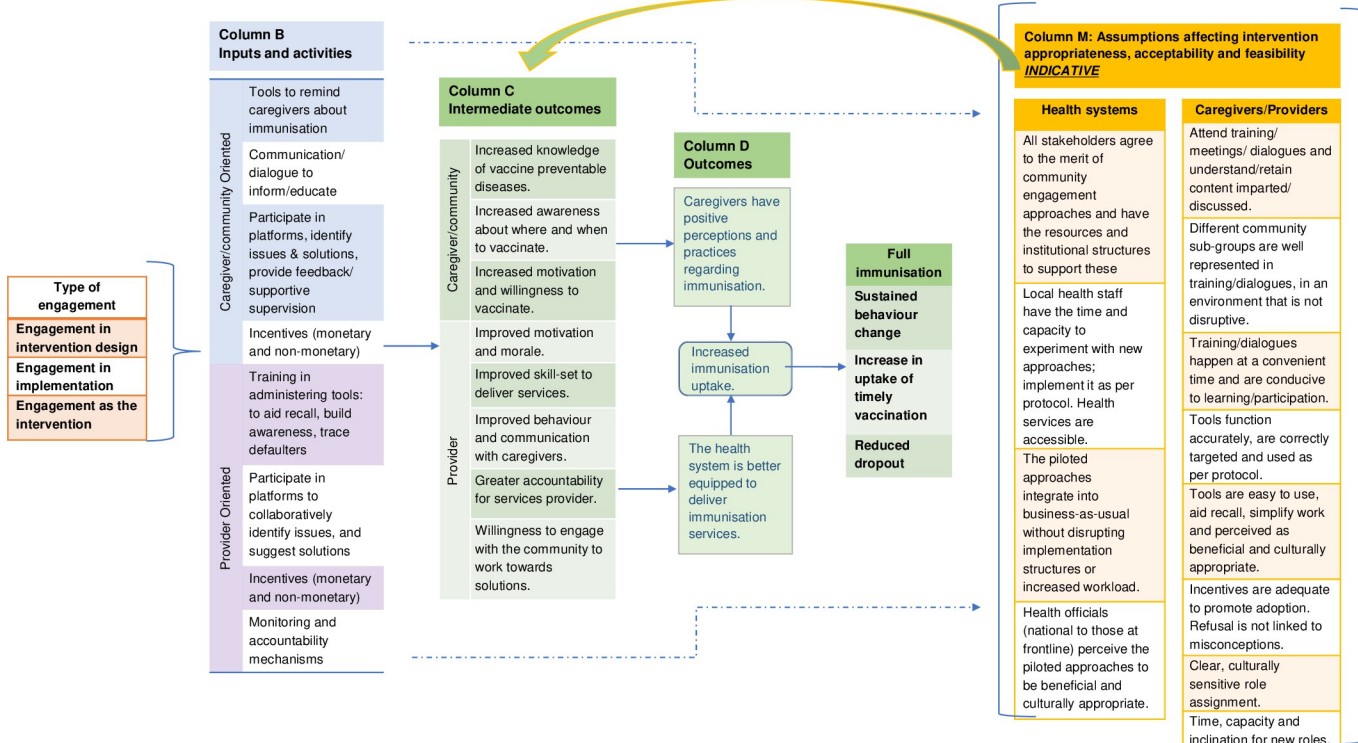

**Fig 1. Theory of change for community engagement-based interventions in immunisation.** Adapted from [17].

*Column A* shows the interventions described above are varying combinations of the following three types:

- **Engagement in intervention design**: the community has input in the design of the intervention, varying from simply being consulted to having some decision-making power to being the final decision-maker.

- **Engagement in intervention implementation:** the community can affect implementation either by providing resources or making decisions such as those around targeting, monitoring, and governance.

- **Engagement as the intervention:** this may include obtaining community buy-in or creation of new community-based structures and cadres of community health workers (CHWs)

CHWs are community members who are trained to provide culturally appropriate health services. Whereas, frontline health workers (FHWs) are typically vaccinators, who also prescribe medicines and/or administer tests.

*Column B* lists tools and approaches interventions used to target caregivers, men's or women's groups, community leaders and health providers.

For inputs and activities to affect the behavioural, social, and practical drivers of immunisation and improve service delivery (*columns C and D)*, the assumptions underlying the causal mechanisms must hold. *Column M* lists the assumptions that likely affected how and whether the interventions were acceptable, appropriate and feasible. However, given the intertwined nature of these implementation outcomes, it can often be difficult to have a discreet set of assumptions that only speak to any one of them. For example, an intervention may be concluded as both inappropriate and unacceptable because it was based on a flawed assumption that no vaccine-related misconceptions exist. The intervention, therefore, neglects misconceptions by design, causing it to be both inappropriate to the context and susceptible to rejection by the participants.

As summarised in S3 Table, studies used mixed methods. They consulted a range of stakeholders to identify potential barriers and enablers that affected the appropriateness, acceptability and feasibility of piloted tools and approaches. The last column provides information on how these interventions map onto the community engagement framework conceptualised by Jain and colleagues [17].

## 3. Results

We present synthesised findings on community-based interventions' appropriateness, feasibility, and acceptability to promote immunisation. We also discuss learnings on formative evaluations as a discipline.

### 3.1. Designing appropriate interventions

Interventions were generally seen as appropriate, although almost all studies tweaked interventions to fit their context better. Appropriateness is seen as the 'perceived fit or relevance of the intervention in a particular setting or for a particular target audience or problem' [14].

**3.1.1 Understanding context and need.** Despite the varied contexts, the studies found the barriers to immunisation uptake to be somewhat similar. Most studies found that caregivers were aware of the benefits of immunisation and that vaccine refusal was not a major barrier to immunisation.

All studies reported strong political will to increase immunisation coverage in the country. Given the global push, the country governments are investing resources in innovative

programmes for last-mile delivery of health services. In Ethiopia, Health Development Army (HDA) was mobilised to link communities with the formal health system. In Nigeria, the government launched the Reaching Every Ward (REW) strategy in 2005 to improve vaccination coverage.

To understand the barriers to immunisation uptake, studies used a combination of tools like extensive baseline surveys, rapid assessments, and social mapping exercises. The baseline surveys revealed a fair understanding of the importance of vaccination in preventing diseases among communities across contexts. In the Collaborative community checklists for immunisation project in Myanmar (CCCI Myanmar), the researchers found no evidence of vaccine refusal and attitudinal barriers were often related to issues of convenience or logistics. In the Vaccine Indicator Reminder project in Pakistan (VIR Pakistan), less than 2% of the respondents thought vaccines to be harmful for children. In the study promoting HDA among pastoralists in Ethiopia (Pastoralists Ethiopia), almost 93% of women believed vaccines prevented diseases. In the Participatory action research project in Nigeria (PAR Nigeria), 88% of respondents in Remo North Local Government Authority recognised the importance of full immunisation. In the Bunza Local Government Authority, however, VIR Nigeria respondents did not demonstrate such high awareness. Knowledge about immunisation schedule and timeliness, however, remained low across contexts. Most respondents were unable to either state the correct age of a child for different vaccines or the number of vaccines to be administered.

Overall, health providers remained an important source for immunisation-related information. In PAR Nigeria, over 90% of the respondents stated health facilities as the main information source. In VIR Nigeria, 46% of respondents expressed an explicit preference for receiving immunisation-related activities from Frontline Health Workers (FHWs). In VIR Pakistan, doctors were the main source of information (46%) after relatives. The mostly commonly discussed supply-side constraints included FHW availability, long wait times, and refusal by FHWs to vaccinate either due to vaccine shortages or there being too few children at the clinic to open a vial.

**3.1.2 Fine-tuning interventions.** Almost all studies revisited intervention modalities to ensure they aligned well with contextual realities and participant preferences.

In VIR Pakistan, the baseline findings showed the need for extensive community engagement to facilitate community acceptance of the reminder bands. This finding informed the design of a comprehensive community engagement plan, which included the identification and creation of a cadre of community health champions to spread awareness about the bands and the need for timely vaccination. In response to caregiver preferences, it was decided that the bands be provided by the FHWs rather than CHWs, which was the original plan. VIR Nigeria also adopted this model to ensure that 'tools were correctly targeted and used as per protocol'.

In CCCI Myanmar, responding to low baseline levels of community knowledge, the intervention scope was expanded to include a health education component to build caregiver understanding of immunisation and equip them to apply checklists as a tool to provide feedback on health service delivery.

In Pastoralists Ethiopia, based on stakeholder and community inputs, traditional leaders ("Abba olla") were assigned a prominent role in organising the HDA network. Their representation in the kebele (village) administration council was also ensured. Other HDA implementation strategies were also tweaked to fit the pastoral context. For instance, the meeting frequency changed from weekly to fortnightly for 1–5 networks and monthly for development group team leaders. Oral narratives were allowed instead of written ones for reporting purposes. Flexibility was introduced in the 1–5 network membership to include less than six community women.

In PAR Nigeria, to prevent elite capture of stakeholder dialogues, the meeting venue was moved from wards to more neutral places to ensure that 'different community sub-groups [were] well represented in training/dialogues, in an environment that is not disruptive'.

## 3.2. Intervention acceptability

Peters et al [14] define acceptability as the 'the perception among stakeholders (for example consumers, providers, managers, policymakers) that an intervention is agreeable'.

**3.2.1 Intervention acceptability by frontline and community health workers.** An important concern with introducing new tools and approaches was that they could add to health worker workload and lead to demotivation or poor performance.

In the Fifth Child Project in Ethiopia (Fifth Child Ethiopia), FHWs felt that the defaulter tracing tool and the Enat Mastawesha calendar made them more efficient and freed up time to reach out to caregivers in remote locations. They eased communication with mothers and the community in general. They also helped better explain the benefits of immunisation. Using the defaulter tracing tool, FHWs successfully immunised 84% of dropout children.

In Pastoralists Ethiopia, health workers at all levels valued women's involvement in the HDA. The FHWs reorganised 1–5 networks when necessitated by group member migration, and this process helped keep the group active.

In CCCI Myanmar, the provider checklist helped increase FHW confidence in providing immunisation services and improved interactions with clients. FHWs admitted to being more polite to caregivers and taking more time to explain things. The FHWs provided earlier notifications and more frequent reminders to community members about the upcoming immunisation sessions.

In PAR Nigeria, health workers acknowledged the importance of the participatory approach that did not 'plan for' but instead 'planned with' community members. Respondents across the board felt that 'all stakeholder groups were actively involved in the PAR process and turnout at meetings was encouraging'. The Principal Medical Officer of Health in one of the wards was termed an 'active change agent'.

VIR studies in Nigeria and Pakistan did not provide detailed insights into intervention acceptability among FHWs. In Nigeria, of the 14 FHWs trained on the use of VIR bands, only five implemented the intervention. In Pakistan, the CHW performance varied across cadres. The Female Community Volunteers of the Polio Eradication Programme participated actively and made referrals to the vaccination centres, and the Lady Health Workers made only 2% of the referrals.

**3.2.2 Intervention acceptability by caregivers and other community members.** The community-engagement tools and approaches were successful in obtaining community buy-in, both among caregivers and community members at large. A key distinguishing aspect of every approach was centre staging the role of community members, particularly leaders, by putting in place systems and structures that ensured their participation in monitoring activities, setting up feedback loops and making immunisation an agenda item in community platforms. In doing so, most interventions either leveraged existing community institutions, like the ward development committees and the social mobilisation committees in PAR Nigeria or the kebele administration in Ethiopia studies, or set up their own platforms, like CCCI Myanmar, to create opportunities for dialogue between service providers and the community to facilitate collective ownership and action.

In Fifth Child Ethiopia, home visits by health workers and Enat Mastawesha calendars improved communication and built trust between caregivers and health providers. Village leaders actively participated in mobilising caregivers and monitoring outreach sessions for

vaccine availability and FHW attendance. At kebele meetings, barriers to immunisation uptake and gaps in health service delivery were discussed to find a way forward.

In Pastoralists Ethiopia, of the 968 women of reproductive age surveyed, 96.3% became a network member. Meeting attendance, however, remained low with only 25% of women attending all or most of the meetings. Additionally, a greater onus was placed on the community to improve immunisation with community leaders, rather than FHWs, reporting on progress at meetings.

In CCCI Myanmar, 81% caregivers credited the checklist for improving their knowledge of immunisation, and 97% expressed willingness to use it again. Ability to provide feedback on health services made them more confident about using these services and more satisfied with the service quality.

Among the VIR studies, in Pakistan, for each DPT vaccine, close to 80% of the children returned to the vaccination centre with the VIR band. In Nigeria, of the 153 respondents at endline, close to 95% mentioned that they would recommend the VIR band to others. The interventions made significant outreach efforts to community influencers to improve band acceptance, though these efforts were better reported in Nigeria than Pakistan.

In PAR Nigeria, a 'majority of the respondents felt that all stakeholder groups were actively involved in the PAR process and turnout for meetings was encouraging'. PAR enabled power-sharing among different groups of stakeholders. Interviews revealed positive participant perceptions about having a voice in the discussions, and overall satisfaction with the decision-making process in the meetings.

## 3.3. Intervention feasibility

While interventions were generally found to be feasible at the scale on which they were implemented, the studies identified key issues around health worker capacity and intervention costs. Feasibility is defined as the 'extent to which an intervention can be carried out in a particular setting or organisation' [14].

**3.3.1 Working with the health systems.** No intervention was delivered as a parallel health service. Integration with the health systems required significant investments in relationship building at the top levels and health staff training at the frontlines. Extensive engagement with stakeholders at all levels of the health systems and trainings for CHWs and FHWs was key in ensuring that piloted interventions fit into the mandates of the health delivery systems. Designing and promoting tools and engagement approaches that eased health worker workload and were perceived as collaborative further helped secure buy in. Though financial incentives were offered to health workers in some contexts, they were limited and their role in motivating health workers was unclear.

The exact details of relationships with health systems varied by context. Burnet institute in Myanmar entered a memorandum of understanding with the Department of Public Health at the Ministry of Health to conduct health-related activities. In CCCI Myanmar and PAR Nigeria, government agency representatives joined the evaluation team as principal investigators, demonstrating the health department's commitment to using evidence to improve immunisation uptake. In Fifth Child Ethiopia and VIR Nigeria and Pakistan, pre-existing relationships with in-country stakeholders was key to building collaborations. IRC in Ethiopia had a long-standing relationship with key stakeholders. In Pakistan, the National Immunization Technical Advisory Group chairperson was also a founding member of the Trust for Vaccines and Immunisation, the implementing agency. This relationship helped secure support from the government Expanded Programme on Immunization (EPI) to pilot VIR bands. In VIR Nigeria, the lead researcher had a long-standing relationship with the Nigeria Primary Health Care

Development Agency, a parastatal body with statutory responsibility of improving access to primary health. In Pastoralists Ethiopia, the study was demand-led, as the Federal Ministry of Health was looking to develop a prototype of the HDA model for the pastoral context.

Buy-in at the top was complemented by significant investments in building the capacity of FHWs and CHWs to roll out the interventions. In the VIR studies, 53 CHWs and 14 FHWs were trained in Nigeria, and in Pakistan 50 participants consisting of Female Community Volunteers, Lady Health Workers and their supervisors, and vaccinators were trained. In Fifth Child Ethiopia, the International Rescue Committee (IRC) conducted a training of trainers for FHWs and their supervisors on the use of the Enat Mastawesha calendar and the defaulter-tracing tool. In CCCI Myanmar, the FHWs and township level supervisors were trained in the use of the provider checklist.

However, the cost implications of these interventions pose a challenge to feasibility. CCCI Myanmar nurtured a new and remunerated cadre of 'checklist assistants' tasked with collating findings from checklists to inform the monthly discussions in the community. VIR Pakistan paid CHWs US$3 for every child referred to a health facility for immunisation. In Nigeria, the CHWs could earn up to US$16 a month for making more than three referrals to a health facility, while FHWs were paid US$13 as an incentive to encourage adherence to the study protocol. Given that these studies did not discuss the cost implications of these intervention components, it is difficult to ascertain how these costs would encroach upon the limited health budgets in L&MIC contexts. In general, cost-related barriers to implementing interventions at scale have not been discussed in most of the studies. However, these barriers require careful consideration when assessing the feasibility of rolling these out through country health systems.

**3.3.2 Improving intervention design and delivery.** The formative evaluations helped highlight design aspects that were critical to consider for intervention feasibility, including community health worker literacy and skill set, financial constraints, and tool design. They also pointed to improvements needed to better adapt interventions to their context. They also highlighted common vulnerabilities in the interventions' theories of change related to participant capacity, institutional readiness, and financial resources that require further investment to enhance intervention feasibility and relevance.

Low literacy levels and limited facilitation skills among health workers posed problems in multiple contexts. In both VIR Nigeria and Pakistan studies, low literacy levels among CHWs affected their ability to complete paperwork to make referrals to health clinics for vaccination, as well as their ability to track and report defaulters. In CCCI Myanmar, checklist assistants required significant handholding to fulfil their role of collating data and communicating findings back to the community. In addition, while the intervention involved project staff conducting monthly health education sessions to build caregiver knowledge and awareness of vaccination, it is not clear how feasible it would be for regular health providers to take over this component. The PAR Nigeria evaluation found that the rural contexts with less cohesion needed stronger leadership for community-based groups. That evaluation pointed to the need to build participant capacity in specific domains such as conflict resolution. From a theory of change perspective, these examples allude to limited capacity among key players to take on new roles and responsibilities, in addition to the potential challenges of implementing similar interventions in low-resource settings.

In CCCI Myanmar, questions about the checklist suggested potential problems with the theory of change. Stakeholders suggested simplifying the community checklist and re-orienting it as a tool that would help the community own its role in immunisation, rather than as 'a tool for critiquing' the health system. The implied lack of clarity on the roles of the stakeholders involved and the key purposes of engagement undermines the intervention theory of change. The underlying distrust of the purpose of the community checklist helps explain why

health providers were reluctant to participate in monthly stocktaking meetings at the community level and work collaboratively to resolve immunisation-related issues, thereby affecting implementation.

Financial constraints also arose as an issue. In PAR Nigeria, some interview subjects reported social mobilisation committees to be 'not so active' in the intervention due to various reasons, including financial constraints. Additionally, the cost of venues was a major cost driver necessitating identifying more cost-effective ways of organising stakeholder dialogues. Besides PAR Nigeria, at least two other studies suggest financial constraints inhibited the smooth implementation of the interventions. In these studies, these constraints affected the organisation of outreach services, which in hard-to-reach populations can be an important deterrent to timely vaccination. Though not discussed by researchers extensively, it may be worth considering how critical components of interventions may be adversely affected by financial barriers.

In addition to the above, the formative evaluations brought forth the need to enhance the functioning and customisation of the tools. For example, in VIR studies in Nigeria and Pakistan, there were serious technical glitches with several bands malfunctioning. In Fifth Child Ethiopia, while the health providers and the community members found the tools useful, suggestions were made to improve them by including pictures of men to encourage male participation in maternal and child health.

## 3.4. Lessons in implementing formative evaluations

**3.4.1 Intervention monitoring.** Systematic monitoring that captures process indicators can point to gaps in the intervention theory of change. The Pastoralist Ethiopia study uses indicators such as trainings and meetings held, groups organised, outreach services set up, supportive supervision visits conducted and so on to track intervention progress. The VIR teams in Nigeria and Pakistan track implementation using tools such as the band tracking register, enrolment summary and attendance registers to name a few. PAR Nigeria developed a checklist to assess implementation of joint actions plans, including meetings held, participant attendance, duration of participation, and achievement of targets set as part of the planning process. In Fifth Child Ethiopia, FHWs maintained a record of outreach sessions conducted, Enat Mastawesha calendars and defaulter-tracing tool distributed, and home visits conducted. The study also reports observing FHW-caregiver interactions to assess adherence to protocol.

Though extremely rich information was collected as part of the monitoring activities in almost all the studies, it was either not reported or reported selectively for only a subset of indicators or intervention components. While it is not clear how the teams determined what monitoring data to report on, systematic reporting, by drawing on existing tools such as the TIDieR intervention reporting guidelines [19], would have further enriched the findings and would have been especially useful to policymakers keen to test the intervention in their context.

**3.4.2 Intrinsic challenges with establishing intervention feasibility.** All interventions were delivered in close collaboration with the existing health machinery, but the researchers and their implementing agency counterparts remained largely responsible for the intervention rollout and delivery. Researchers and their counterparts' roles included sensitisation and awareness generation among community stakeholders, capacity building of CHWs and FHWs, setting up systems and protocols for intervention delivery, collecting monitoring data and providing supportive supervision. In both the Ethiopia studies, the study teams were involved in logistic support such as transporting vaccines, ensuring cold chain functionality, and ensuring EPI systems remained functional. In CCCI Myanmar, researchers stepped in to compensate for capacity shortfalls among the checklist assistants. In PAR Nigeria, the research team helped resolve conflicts at various dialogues to facilitate consensus on important service delivery issues.

While the very raison d'etre of formative evaluations is to allow an iterative approach to establish proof-of-concept, these roles raise concerns about the health systems' commitment and capacity to deliver the interventions without support from research teams. The existing literature does not shed light on how a formative evaluation can address the seemingly paradoxical expectation of establishing a proof-of-concept with researcher involvement while testing for its appropriateness, acceptability and feasibility from a service provider perspective.

## 4. Discussion

The overall agreement in the findings that vaccine hesitancy was not a major cause of low immunisation coverage is consistent with other recent studies [20]; however, it may be masking underlying variability between contexts and reporting systems. Studies did not use a core set of indicators systematically to explore factors such as caregiver knowledge and awareness. For example, while adverse events following immunisation were reported as an issue in Nigeria, it is unclear if this was absent in other contexts or simply was not explored.

The findings about the success of community engagement tools and approaches at obtaining buy-in among caregivers and community members at large also resonate with the literature on community engagement and its importance in designing, planning and implementing interventions so that they are more likely to be acceptable to the community, feasible, and appropriate for the context [17, 21, 22]. The interventions in the synthesis did so by assigning roles to community leaders in programme monitoring and leveraging on existing community platforms to facilitate shared accountability between the community and FHWs.

Appropriateness, acceptability, and feasibility of interventions hinge on a few key aspects. At the outset, there needs to be political will and commitment to address public health policy such as issues of immunisation [23]. These were demonstrated in all country contexts and helped the teams get buy-in for piloting community-based interventions. Pre-existing relationships between researchers or their implementing agency counterparts and country governments (VIR studies and Fifth Child Ethiopia) and the ability to identify and create champions within the government (CCCI Myanmar and PAR Nigeria) also mattered. Embedding community-engagement approaches into the existing health system and extensive health worker training further helped in intervention delivery.

However, from a provider perspective, intervention feasibility is mediated by a few other important factors, including health worker capacity, a lack of cost data, and the limits of the overall health system. Whether local capacity is sufficient to deliver these interventions in low resource L&MIC contexts remains a concern [17, 24].

That said, the iteration and reflexivity that formative evaluations encouraged ensured that there was constant re-alignment of intervention strategies and processes such that they better addressed the underlying theoretical assumptions of the interventions to encourage uptake of immunisation services. Learnings at baseline and through the course of the study were used to strengthen community engagement in design and implementation, drawing on improved contextual understanding to enhance intervention acceptability, feasibility, and appropriateness.

## 5. Limitations

The synthesis is an overview of what different studies find about the appropriateness, acceptability, or feasibility of the interventions they evaluated. Pooling data from different studies across outcomes to provide an aggregate estimate was neither desirable nor feasible in the absence of standardised measures for key outcomes.

Additionally, our findings and interpretations are bound by the limitations of our data sources. As such, this synthesis is not a comprehensive account of all activities undertaken by

the study teams, but those that were reported to 3ie. The quality of reporting also varied from team to team. Some teams provided detailed logs of activities carried out during the reporting period, highlighting any risks and challenges faced, while others only listed activities with little to no reflection or insights on implementation challenges.

## Supporting information

**S1 Table. Formative evaluation coding structure.**
(DOCX)

**S2 Table. Description of interventions.**
(DOCX)

**S3 Table. Overview of formative evaluations included in the synthesis.**
(DOCX)

## Acknowledgments

We would like to thank Molly Abbruzzese and Sohail Agha, former Senior Program Officers, Gates Foundation, Seattle, for their guidance, important inputs and support on this review. The authors would like to thank Sapna Desai, Ritwik Sarkar, and Paul Thissen for providing useful inputs and insights that helped to improve the quality of the article. Shradha Parsekar and Tanvi provided editorial support that helped with the submission process. We would further like to thank Marie Gaarder, Executive Director, 3ie, for her guidance on the review process. We also want to thank Radhika Menon and Ananta Seth for quality assuring the formative evaluations as a part of the 3ie team involved in the immunisation evidence programme.

## Author Contributions

**Conceptualization:** Stuti Tripathi, Monica Jain.

**Formal analysis:** Stuti Tripathi.

**Methodology:** Stuti Tripathi, Monica Jain, Avantika Bagai, Kirthi V. Rao.

**Resources:** Avantika Bagai.

**Visualization:** Avantika Bagai.

**Writing – original draft:** Stuti Tripathi, Monica Jain, Avantika Bagai.

**Writing – review & editing:** Stuti Tripathi, Monica Jain, Avantika Bagai, Kirthi V. Rao.

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
