## [Decision Letter · Decision Letter 0]

14 Jun 2022

PONE-D-21-37729Designing appropriate, acceptable and feasible community-engagement approaches to improve routine immunisation outcomes in low- and middle-income countries: a synthesis of 3ie-supported formative evaluationsPLOS ONE

Dear Dr. Tripathi,

Thank you for submitting your manuscript to PLOS ONE. After careful consideration, we feel that it has merit but does not fully meet PLOS ONE’s publication criteria as it currently stands. Therefore, we invite you to submit a revised version of the manuscript that addresses the points raised during the review process.

We look forward to receiving your revised manuscript.

Kind regards,

Dylan A Mordaunt, MD, MPH, FRACP

Academic Editor

PLOS ONE

**Journal requirements:**

“This research has been undertaken as a part of 3ie’s immunisation evidence programme, supported by the Bill and Melinda Gates Foundation, Seattle, USA. We would like to thank Sohail Agha, Senior Program Officer, Gates Foundation, Seattle, for his continued support and engagement on the evidence programme and the review. Thanks to Molly Abbruzzese, whose guidance helped shape the scope of our evidence programme and aided the conceptualisation of this review. The authors would like to thank Sapna Desai, Ritwik Sarkar, and Paul Thissen for providing useful inputs and insights that helped to improve the quality of the article. Shradha Parsekar and Tanvi provided editorial support that helped with the submission process. We would further like to thank Marie Gaarder, Executive Director, 3ie, for her guidance on the review process. We also want to thank Radhika Menon and Ananta Seth for quality assuring the formative evaluations as a part of the 3ie team involved in the immunisation evidence programme.

“This work was made possible by a grant from the Bill & Melinda Gates Foundation to the International Initiative for Impact Evaluation (3ie). The grant number is OPP1115129.

Website URL - https://www.gatesfoundation.org/

Under the grant conditions of the Foundation, a Creative Commons Attribution 4.0 Generic License has already been assigned to the Author Accepted Manuscript version that might arise from this submission.

“I have read the journal's policy and the authors of this manuscript have the following competing interests: Supported by the Bill and Melinda Gates Foundation, the International Initiative for Impact Evaluation (3ie) provided funding and technical assistance for the six formative evaluations included in this synthesis. 3ie’s technical assistance included review of study designs, analysis plans, and data collection instruments; and ongoing quality assurance, including advising research teams on various study aspects such as challenges in study implementation and engagement with stakeholders to promote uptake and use of evidence.

Authors MJ and AB were involved in reviewing proposals for formative evaluation study grants and providing research teams with technical assistance. However, procedural safeguards were put in place to mitigate any risk of conflicting or competing interest. First, the lead author of this study, ST, has not been involved in quality assurance of these formative evaluations. Second, the review was undertaken only after all formative evaluation reports were completed and made publicly available. Lastly, the authors have no financial interest in this area.”

6. We note that you have referenced (ain M, Shisler S, Lane C, Bagai A, Brown E, Engelbert M, et al. Use of community engagement interventions to improve child immunisation in low‐ and middle‐income countries: a systematic review and meta‐analysis International Development Coordinating Group at The Campbell Collaboration. 2021; Unpublished results) which has currently not yet been accepted for publication. Please remove this from your References and amend this to state in the body of your manuscript as detailed online in our guide for authors

7. Please include a copy of Tables 1-3 which you refer to in your manuscript.

**Additional Editor Comments:**

Thank you for your submission. From my perspective the main issue is a slight lack of clarity in description of the methods. Although this doesn't fit clearly with one or other type of study type, it would be worth considering whether either the SRWR or COREQ reporting guideliens would beo worth following- https://www.equator-network.org/?post_type=eq_guidelines&eq_guidelines_study_design=qualitative-research&eq_guidelines_clinical_specialty=0&eq_guidelines_report_section=0&s=.

With regards to the critiera for publication:

1. The study appears to present the results of original research.

2. Results reported do not appear to have been published elsewhere.

3. Experiments, statistics, and other analyses are performed to a high technical standard but require some clarity on methods.

4. Conclusions are presented in an appropriate fashion and are supported by the data.

5. The article is presented in an intelligible fashion and is written in standard English.

6. The authors should clarify whether an IRB approval was required or saught in their jurisdiction.

7. The article would benefit from review with regards to structured reporting requirements for qualitative research.

Reviewers' comments:

Reviewer's Responses to Questions

**Comments to the Author**

1. Is the manuscript technically sound, and do the data support the conclusions?

Reviewer #1: Yes

2. Has the statistical analysis been performed appropriately and rigorously? 

Reviewer #1: N/A

3. Have the authors made all data underlying the findings in their manuscript fully available?

Reviewer #1: No

4. Is the manuscript presented in an intelligible fashion and written in standard English?

Reviewer #1: Yes

5. Review Comments to the Author

Reviewer #1: Reviewer’s report: BM

Summary: Thank you for the opportunity to peer review this important article. The article, ‘Designing, appropriate, acceptable, and feasible community-engagement approaches to improve routine immunisation outcomes in low- and middle-income countries: a synthesis of 3ie-supported formative evaluations.’ Being a systematic review utilising inductive inquiry, this is in a way secondary data study. These results will contribute to the body of knowledge on correlates for demand of vaccination service, particularly in the area of community engagement.

Overall impression: The aim of the study was to synthesize qualitative findings from the six (6) 3ie-supported formative evaluations undertaken in Ethiopia, Myanmar, Nigeria, and Pakistan by International Initiative for Impact Evaluation (3ie) organisation in-order to:

- broaden the understanding of formative evaluations

- better understand community influencers

Major issues:

• Positive note: With the use of Critical Appraisal Skills Programme (CASP) checklist tool the authors followed the research methodology clearly and the documentation is well presented as a qualitative research with clear results, including documentation on study ethics. Formative evaluation structure is documented, and summary description of results has been illustrated in tables 2 and 3 of the supplementary material.

Minor issues:

Line 103: requires editing

Line 127: where is this figure 1 (was it uploaded separately)

Recommendation: • Accept article with minor changes/update/edits.

6. PLOS authors have the option to publish the peer review history of their article (what does this mean?). If published, this will include your full peer review and any attached files.

Reviewer #1: No

---

## [Author Response · Author response to Decision Letter 0]

8 Sep 2022

As mentioned above, we were able to incorporate all the suggested revisions as detailed in the document titled Response to reviewers (pasted below). In the Response to reviewers document, we have followed the order in which the comments were shared and provided responses for each in the second column. As suggested, we have appended the revised funding statement and the revised statement of competing interests in the cover letter. 

RESPONSE: We have revised the manuscript to ensure it meets PLOS ONE's style requirements. 

2 Acknowledgement and Funding statement declaration

“This research has been undertaken as a part of 3ie’s immunisation evidence programme, supported by the Bill and Melinda Gates Foundation, Seattle, USA. We would like to thank Sohail Agha, Senior Program Officer, Gates Foundation, Seattle, for his continued support and engagement on the evidence programme and the review. Thanks to Molly Abbruzzese, whose guidance helped shape the scope of our evidence programme and aided the conceptualisation of this review. The authors would like to thank Sapna Desai, Ritwik Sarkar, and Paul Thissen for providing useful inputs and insights that helped to improve the quality of the article. Shradha Parsekar and Tanvi provided editorial support that helped with the submission process. We would further like to thank Marie Gaarder, Executive Director, 3ie, for her guidance on the review process. We also want to thank Radhika Menon and Ananta Seth for quality assuring the formative evaluations as a part of the 3ie team involved in the immunisation evidence programme.

Please remove any funding-related text from the manuscript and let us know how you would like to update your Funding Statement. 

Currently, your Funding Statement reads as follows:

“This work was made possible by a grant from the Bill & Melinda Gates Foundation to the International Initiative for Impact Evaluation (3ie). The grant number is OPP1115129.

Website URL - https://www.gatesfoundation.org/

Under the grant conditions of the Foundation, a Creative Commons Attribution 4.0 Generic License has already been assigned to the Author Accepted Manuscript version that might arise from this submission.

RESPONSE: We have revised the Acknowledgement in the manuscript (please see the changes in the Revised article with changes highlighted). We have also revised the Funding statement declaration and included it in the Cover letter as suggested. 

3 COMPETING INTERESTS

“I have read the journal's policy and the authors of this manuscript have the following competing interests: Supported by the Bill and Melinda Gates Foundation, the International Initiative for Impact Evaluation (3ie) provided funding and technical assistance for the six formative evaluations included in this synthesis. 3ie’s technical assistance included review of study designs, analysis plans, and data collection instruments; and ongoing quality assurance, including advising research teams on various study aspects such as challenges in study implementation and engagement with stakeholders to promote uptake and use of evidence.

Authors MJ and AB were involved in reviewing proposals for formative evaluation study grants and providing research teams with technical assistance. However, procedural safeguards were put in place to mitigate any risk of conflicting or competing interest. First, the lead author of this study, ST, has not been involved in quality assurance of these formative evaluations. Second, the review was undertaken only after all formative evaluation reports were completed and made publicly available. Lastly, the authors have no financial interest in this area.”

RESPONSE: As recommended, we have appended the new Competing interests statement in the Cover letter. 

4 We note that you have indicated that data from this study are available upon request. PLOS only allows data to be available upon request if there are legal or ethical restrictions on sharing data publicly. For more information on unacceptable data access restrictions, please see 

http://journals.plos.org/plosone/s/data-availability#loc-unacceptable-data-access-restrictions.

b) If there are no restrictions, please upload the minimal anonymized data set necessary to replicate your study findings as either Supporting Information files or to a stable, public repository and provide us with the relevant URLs, DOIs, or accession numbers. 

For a list of acceptable repositories, please see 

http://journals.plos.org/plosone/s/data-availability#loc-recommended-repositories.

RESPONSE: We have appended an amended Data availability statement in the Cover letter, highlighting where information is not publicly available and reasons for the same. 

5 Please include your full ethics statement in the ‘Methods’ section of your manuscript file. In your statement, please include the full name of the IRB or ethics committee who approved or waived your study, as well as whether or not you obtained informed written or verbal consent. If consent was waived for your study, please include this information in your statement as well.

RESPONSE: We have revised the manuscript Methods section significantly to include information on ethics approval. Please see the manuscript with changes highlighted for details. 

6 We note that you have referenced (Jain M, Shisler S, Lane C, Bagai A, Brown E, Engelbert M, et al. Use of community engagement interventions to improve child immunisation in low‐ and middle‐income countries: a systematic review and meta‐analysis International Development Coordinating Group at The Campbell Collaboration. 2021; Unpublished results) which has currently not yet been accepted for publication. Please remove this from your References and amend this to state in the body of your manuscript as detailed online in our guide for authors: http://journals.plos.org/plosone/s/submission-guidelines#loc-reference-style

RESPONSE: The then unpublished report mentioned by the reviewer was published on 27 July 2022. Hence, we have updated the reference to the following (reference no. 17): 

Jain, M., Shisler, S., Lane, C., Bagai, A., Brown, E., Engelbert, M., Vardy, Y., Eyers, J., Leon, D. A., & Parsekar, S. S. (2022). Use of community engagement interventions to improve child immunisation in low- and middle-income countries: A systematic review and meta-analysis. Campbell Systematic Reviews, 18, https://doi.org/10.1002/cl2.1253

7 Please include a copy of Tables 1-3 which you refer to in your manuscript.

RESPONSE: 

Re Tables 1-3

In keeping with formatting and file naming guidelines, we have uploaded Tables 1-3 as separate supporting information tables (S1 Table, S2 Table and S3 Table) and cited them in-text accordingly in the revised manuscript. 

Changes to the reference list: 

We changed one reference (previously 15 and currently no.17) to reflect publication status. Two references were added in the methods section in response to the additional editor comment below (currently reference numbers 15 and 16). 

One reference was removed from the list and in text since we agreed it was not required (previously 25).

All the other references are confirmed to be correct. 

Please note that reference 25 below is cited in S2 Table. 

World Health Organization. Immunization in practice: a practical guide for health staff – 2015 update. Geneva: World Health Organization; 2015.

8 (Additional editor comments) Thank you for your submission. From my perspective the main issue is a slight lack of clarity in description of the methods. Although this doesn't fit clearly with one or other type of study type, it would be worth considering whether either the SRQR or COREQ reporting guidelines would be worth following- https://www.equator-network.org/?post_type=eq_guidelines&eq_guidelines_study_design=qualitative-research&eq_guidelines_clinical_specialty=0&eq_guidelines_report_section=0&s=.

With regards to the critiera for publication:

1. The study appears to present the results of original research.

2. Results reported do not appear to have been published elsewhere.

3. Experiments, statistics, and other analyses are performed to a high technical standard but require some clarity on methods.

4. Conclusions are presented in an appropriate fashion and are supported by the data.

5. The article is presented in an intelligible fashion and is written in standard English.

6. The authors should clarify whether an IRB approval was required or sought in their jurisdiction.

7. The article would benefit from review with regards to structured reporting requirements for qualitative research.

RESPONSE: We have updated the Methods section as well as the References accordingly. 

9 [Reviewers' comments: Reviewer's Responses to Questions Comments to the Author] Summary: Thank you for the opportunity to peer review this important article. The article, ‘Designing, appropriate, acceptable, and feasible community-engagement approaches to improve routine immunisation outcomes in low- and middle-income countries: a synthesis of 3ie-supported formative evaluations.’ Being a systematic review utilising inductive inquiry, this is in a way secondary data study. These results will contribute to the body of knowledge on correlates for demand of vaccination service, particularly in the area of community engagement.

Overall impression: The aim of the study was to synthesize qualitative findings from the six (6) 3ie-supported formative evaluations undertaken in Ethiopia, Myanmar, Nigeria, and Pakistan by International Initiative for Impact Evaluation (3ie) organisation in-order to:

- broaden the understanding of formative evaluations

- better understand community influencers

Major issues:

• Positive note: With the use of Critical Appraisal Skills Programme (CASP) checklist tool the authors followed the research methodology clearly and the documentation is well presented as a qualitative research with clear results, including documentation on study ethics. Formative evaluation structure is documented, and summary description of results has been illustrated in tables 2 and 3 of the supplementary material.

Minor issues:

Line 103: requires editing

Line 127: where is this figure 1 (was it uploaded separately)

Recommendation: • Accept article with minor changes/update/edits.

RESPONSE: Thank you for your comments. We have addressed the minor comments in the revised manuscript and correctly added Fig 1. We had uploaded it separately earlier but it was not correctly placed in the manuscript. 

10 While revising your submission, please upload your figure files to the Preflight Analysis and Conversion Engine (PACE) digital diagnostic tool, https://pacev2.apexcovantage.com/. 

PACE helps ensure that figures meet PLOS requirements. To use PACE, you must first register as a user. Registration is free. Then, login and navigate to the UPLOAD tab, where you will find detailed instructions on how to use the tool. If you encounter any issues or have any questions when using PACE, please email PLOS at figures@plos.org. Please note that Supporting Information files do not need this step. 

RESPONSE: Thanks for pointing us to the PACE tool and helping us to upload the compliant figure correctly per journal requirements.

---

## [Editor Report · Decision Letter 1]

13 Sep 2022

Designing appropriate, acceptable and feasible community-engagement approaches to improve routine immunisation outcomes in low- and middle-income countries: a synthesis of 3ie-supported formative evaluations

PONE-D-21-37729R1

Dear Dr. Tripathi,

We’re pleased to inform you that your manuscript has been judged scientifically suitable for publication and will be formally accepted for publication once it meets all outstanding technical requirements.

Kind regards,

Dylan A Mordaunt, MD, MPH, FRACP

Academic Editor

PLOS ONE

Additional Editor Comments (optional):

Thank you for your resubmission. This now meets the criteria for publication.
---

## [Editor Report · Acceptance letter]

19 Sep 2022

PONE-D-21-37729R1 

Designing appropriate, acceptable and feasible community-engagement approaches to improve routine immunisation outcomes in low- and middle-income countries: a synthesis of 3ie-supported formative evaluations 

Dear Dr. Tripathi:

I'm pleased to inform you that your manuscript has been deemed suitable for publication in PLOS ONE. Congratulations! Your manuscript is now with our production department. 

Kind regards, 

on behalf of

Associate Professor Dylan A Mordaunt 

Academic Editor

PLOS ONE